# The Synthesis of Biodegradable Poly(L-Lactic Acid)-Polyethylene Glycols Copolymer/Montmorillonite Nanocomposites and Analysis of the Crystallization Properties

**Jiunn-Jer Hwang [1,2,\*], Su-Mei Huang [1], Wen-Yang Lin [3] , Hsin-Jiant Liu [4], Cheng-Chan Chuang [4] and Wen-Hui Chiu [5]**

1    Department of Chemical Engineering, Army Academy, Taoyuan City 32092, Taiwan; huangsumei888@hotmail.com

2    Center for General Education, Chung Yuan Christian University, Taoyuan City 32023, Taiwan

3    Department of Biological Science and Technology, National Chiao Tung University, Hsin Chu 30010, Taiwan; gasboy1121@gmail.com

4    Department of Chemical and Materials Engineering, Vanung University, Taoyuan City 32061, Taiwan; geneliu329@gmail.com (H.-J.L.); nanoca41@gmail.com (C.-C.C.)

5    Department of Cosmetic Science, Vanung University, Taoyuan City 32061, Taiwan; whc@mail.vnu.edu.tw

\*    Correspondence: jiunnjer1@hotmail.com; Tel.: +886-3-466-4600 (ext. 345420)

**Abstract:** This study makes use of polycondensation to produce poly (L-lactic acid)-(polyethylene glycols), a biodegradable copolymer, then puts it with organically modified montmorillonite (o-MMT) going through an intercalation process to produce a series of nanocomposites of PLLA-PEG/o-MMT. The exfoliation and intercalation of the montmorillonite-layered structure could be found through X-ray diffraction and transmission electron microscopy. The lower the molecular weight of poly (ethylene glycol), the more obvious the exfoliation and dispersion. The nanocomposites were investigated under non-isothermal crystallization and isothermal crystallization separately via differential scanning calorimetry (DSC). After the adding of o-MMT to PLLA-PEG copolymers, it was found that the PLLA-PEG nanocomposites crystallized slowly and the crystallization peak tended to become broader during the non-isothermal crystallization process. Furthermore, the thermal curve of the non-isothermal melt crystallization process of PLLA-PEG copolymers with different proportions of o-MMT showed that the melting point decreased gradually with the increase of o-MMT content. In the measurement of isothermal crystallization, increasing the o-MMT of the PLLA-PEG copolymers would increase the $t_{1/2}$ (crystallization half time) for crystallization and decrease the value of $\Delta Hc$. However, the present study results suggest that adding o-MMT could affect the crystallization rate of PLLA-PEG copolymers. The o-MMT silicate layer was uniformly dispersed in the PLLA-PEG copolymers, forming a nucleating agent. The crystallization rate and the regularity of the crystals changed with the increase of the o-MMT content, which further affected the crystallization enthalpies.

**Keywords:** poly (L-lactic acid); polyethylene glycols; montmorillonite; nanocomposites; crystallization properties

## 1. Introduction

As there is increasing awareness on ecology and the environment, the United Nations Commission on Sustainable Development proposed to "replace non-recyclable resources with recyclable ones" [1]. The customary procedure of producing plastic materials leads to global warming, and un-degraded plastic waste is "ingested" to marine animals [2]. Therefore, it is crucial to develop novel recyclable and degradable materials (called green earth materials) to recover the global environment [3,4].

Poly (lactic acid) (PLA) is a hydrophilic aliphatic polyester base on the hydroxyl group –OH, and hydrocarbon, which belong to one of the rising biodegradable green composites [5–7]. Biodegradable green composites are eco-friendly, can be decomposed,

and are applicable in biomedicine [8,9]. Therefore, scientists will gradually replace the plastic materials refined from petroleum-based chemicals with PLA [10–12]. PLA has low immunogenicity, satisfactory biocompatibility, and excellent mechanical properties, it is widely applied in medical treatments, such as surgical sutures, bone nails, bone plates, fixed frameworks, carriers of pharmaceutical deliveries, temporary matrices, and scaffolds in tissue engineering [13,14]. However, the poor flexibility of PLA limits its further application [15–20]. It was discovered that L-lactic acid-formed poly (L-lactic acid) (PLLA) has lower crystallinity, and a softer, much higher biological decomposition effect than general PLA. Therefore, PLLA has been replaced by PLA in the application of biomedicine [21–23]. PLLA has poor transparency, some weaknesses, and quickly breaks down during processing and molding.

Additionally, poly (ethylene glycol) contains good physical properties and chemical properties, perfect hydrophilicity, thermoplasticity, and is non-toxic. Moreover, it could dissolve in water and several organic solvents, an ideal composite used in biomaterial science and medical application fields [24–26]. Therefore, researchers began developing a unique function of blend materials, called diblock copolymer of PLLA-PEG, to extend related applications [27–33].

In this study, PLLA-PEG copolymers were generated by co-polymerization of 3,6-dimethyl-1,4-dioxane-2,5-dione and poly (ethylene glycol) as the main components of the material, aimed at improving the stiffness and flexibility of PLLA, and reducing the defect rate of the product during processing by introducing the low molecular weight of PEG. Subsequently, a series of non-toxic biodegradable PLLA-PEG/o-MMT copolymer nanocomposites were prepared by adding different organically modified montmorillonite (o-MMT) proportions. The effects of adding various ratios of o-MMT on its physical properties, crystallization pattern, and crystallization growth rate were investigated. It is expected that the PLLA-PEG nanocomposites with the addition of o-MMT can improve the crystallization property of the PLLA-PEG and control the processing temperature. In addition, the o-MMT controlled crystallization degree and the crystallization temperature were measured. The influence of the size of the PEG molecular chain on the morphology and related crystalline properties of the PLLA-PEG nanocomposites is also discussed.

## 2. Experimental

### 2.1. Materials

Materials: 3,6-Dimethyl-1,4-dioxane-2,5-dione (DL-Lactide) (Bio Invigor Corporation, Taipei, Taiwan); Poly (ethylene glycol), PEG-200, PEG-1000, PEG-2000 (SHOWA Chemical Co., Tokyo, Japan); Chloroform (TEDIA Chemicals, Fairfield, OH, USA); PK805 montmorillonite (Pai Kong nano technology Co., LTD., Taipei, Taiwan); dilauryl dimethyl ammonium bromide, ($[CH_3(CH_2)_{11}]_2(CH_3)_2NBr$), (TCI Chemicals, Portland, OR, USA).

### 2.2. Preparation of the Organophilic Montmorillonite

The proper amount of montmorillonite was in distilled water (montmorillonite: $H_2O$ = 1:100) for 12 h. The scaled proper amount of surfactant ($[CH_3(CH_2)_{11}]_2(CH_3)_2NBr$) was dissolved in distilled water, where we added a couple of drops of hydrochloric acid inside. Two solutions were mixed for 12 h. The usage formula: (surfactant usage = CEC (montmorillonite cation exchange capacity: 98 meq/100 g) $\times$ surfactant molecular weight $\times$ montmorillonite weight $\times$ 1.2 $\times$ $10^{-3}$). Next, the organically modified MMT was filtered, and then washed with distilled water, and the filtered water was titrated with silver nitrate solution until noyellow AgBr precipitation appeared. The modified MMT was dried and triturated in a 100 °C vacuum oven, and then the dried o-MMT powder was sifted through a 325-mesh strainer to get the particle size to about 40 μm. Finally, X-ray diffraction (XRD, Vanung University Material Measurement Center, Waltham, MA, USA) was used to identify successful products of MMT.

*2.3. Synthesis of Copolymers and Preparation of Composite Materials*

2.3.1. Synthesis of Copolymers

We placed DL-Lactide and polyethylene glycols with a molar ratio of 100/1 into a reaction vessel, and then filled it with nitrogen to keep the reaction system operating. We added a catalyst to the polymerization, to remain under 120–130 °C for 24 h. We proceeded with purification and desiccation of the copolymer in a drying cabinet.

2.3.2. Preparation of PLLA-PEG Copolymer Nanocomposites

We scaled 3g PLLA-PEG copolymer and added 0.015, 0.03, and 0.045 g of modified o-MMT, separately, to dissolve in 20 mL of chloroform, and then stirred it for 24 h. We decanted the mixed well copolymer solution into a Teflon disk, and then put it into a 60 °C vacuum oven for 24 h to remove chloroform entirely; we prepared film forming at the last step. The prepared composite material should be kept in a drying cabinet to prevent hydrolysis, to avoid affecting the property of composite material.

*2.4. Characterizations*

2.4.1. X-ray Diffraction (XRD)

Thermo Electron ARL X'TRAS, Cu Kα radiation: λ = 1.541 Å. The interlayer distance (d) was calculated according to Bragg's equation (nλ = 2d sinθ). To analyze the PLLA-PEG copolymer nanocomposite purpose, the setting of the scanning range (2θ) was within 2–10°, and the scanning speed was 0.04°/min.

2.4.2. Gel Permeation Chromatography (GPC)

Molecular weight and polydispersity index(Mw/Mn) of copolymers were measured on a Waters GPC, consisting of a Waters 1515 isocratic HPLC pump, 2414 RI detector (Vanung University Material Measurement Center, Milford, CT, USA), and two PL-gel mix C columns at 30 °C. Tetrahydrofuran was used as the mobile phase at a flow rate of 1.0 mL/min, and the monodisperse polystyrenes used a standard for calibration.

2.4.3. Fourier Transform Infrared Spectrometer (FT-IR)

Perkin Elmer Spectrum One (Vanung University Material Measurement Center, Akron, OH, USA), experimental conditions: scan range 4000 cm$^{-1}$~650 cm$^{-1}$, ZnSe MIRacle ATR Accessory, resolution 4 cm$^{-1}$.

2.4.4. $^1$H-NMR Spectrometer

Model: Bruker 500 MHz NMR (NTU Precious Instrument Center, Ettlingen, Germany), experimental conditions: sample compounding concentration of 10 mg/mL d-chloroform.

2.4.5. Transmission Electron Microscope (TEM)

The samples for TEM analyses were prepared by placing the films of PLLA-PEG copolymer nanocomposites into epoxy resin capsules, and by curing these capsules at 70 °C for 24 h in a vacuum oven. Then, the cured epoxy resin that contained the PLLA-PEG copolymer nanocomposites was microtomed with a Reichert-Jung Ultracut-E to form 60–90 nm-thick slices (Chung Yuan Christian University Precious Instrument Center, Optische Werke AG Wien, Chungli, Taiwan). Subsequently, one layer of carbon around 10 nm thick was deposited onto the slices, which were placed on 100-mesh copper nets for TEM observation using a JEOL 2010 instrument (Tokyo, Japan), operated at an accelerating voltage of 200 kV.

2.4.6. Non-Isothermal and Isothermal Crystallization Experiments

We proceeded with non-isothermal and isothermal crystallization experiments using a Perkin-Elmer Pyris 6 Differential Scanning Calorimeter (DSC, Vanung University Material Measurement Center, Akron, OH, USA) [34,35]. We conducted dual-point adjustment

through a standardized test block of indium (mp = 156.60 °C) and zinc (mp = 419.47 °C) before experimenting, and then circulated with 20 mL/min flow speed high purity nitrogen.

## 3. Results and Discussion

### 3.1. PLLA-PEG Copolymer Synthesis and Characterization

We placed the sample tube into an exclusive NMR tube to offset the line broadening impact, which originated from both dipolar interactions (Hdd) and chemical shift anisotropy (HCS) via high-speed spinning. Furthermore, the deuterium lock and shimming method promoted the magnetic field homogeneity to get a high-resolution spectrum. As presented in Figure 1, segment z was the main structure of PLLA, the H atom's chemical shift (δ) of carbon at label (a), and the H atom's chemical shift (δ) of $CH_2$ at label (d) in segment y showed between 5.15 and 5.23 ppm. x was the main structure of PEG, the H atom's chemical shift (δ) of $CH_2$ at label (c) showed 3.65 ppm. At the z segment, the H atom's chemical shift (δ) of -$CH_3$ at label (b) appeared "multiple", between 1.4 and 1.6 ppm.

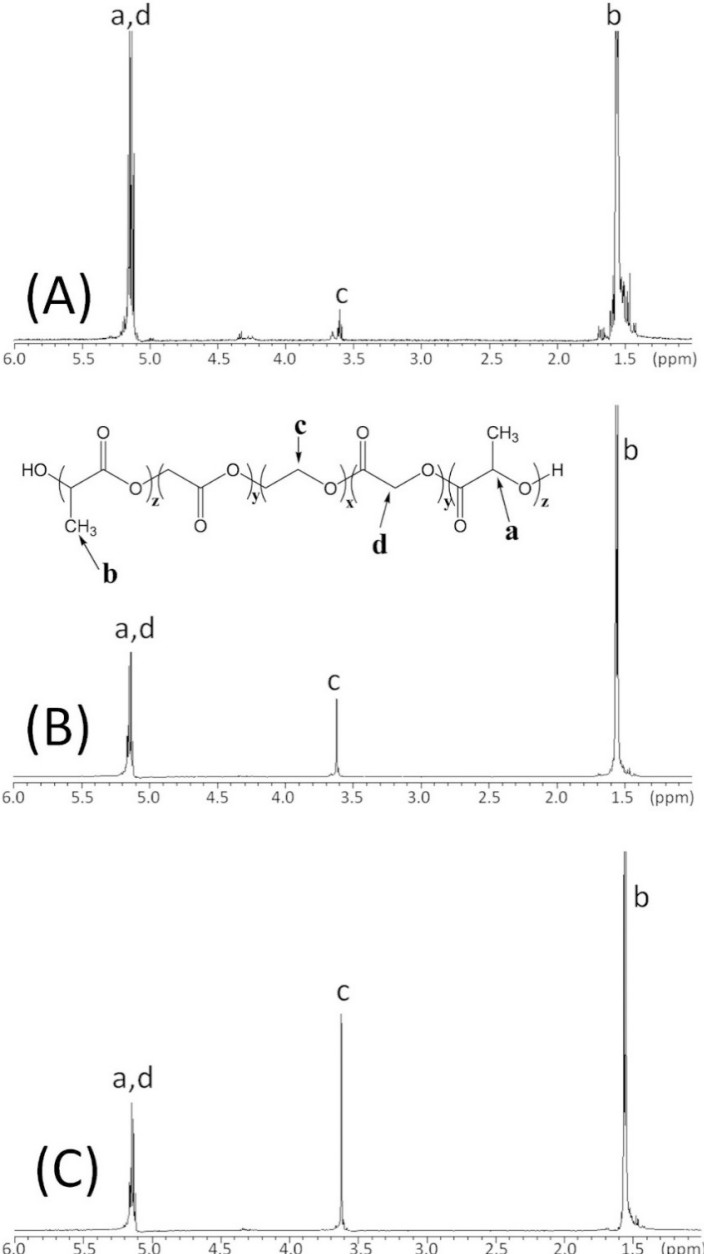

**Figure 1.** $^1$H-NMR spectra of (**A**) PLLA-PEG200; (**B**) PLLA-PEG1000; (**C**) PLLA-PEG2000.

Figure 2a–c indicate PEG1000, PLLA, and PLLA-PEG1000 infrared spectroscopy, respectively. The C=O stretching vibrational absorption of the ester functional group of PLLA appeared at 1757 cm$^{-1}$; -CH$_3$ stretching vibrational absorption of PEG showed between 3000 and 2800 cm$^{-1}$. According to Figure 2c, PLLA-PEG1000 infrared spectroscopy and $^1$H-NMR spectroscopy could figure out that the series of syntheses of the PLLA-PEG block copolymer was successful. Moreover, following the frequency of the 1H-NMR quantity, the relative abundance ratio would be found out: PLLA and PEG200 with a ratio of 12/1, PLLA and PEG1000 with a ratio 8/1, and PLLA and PEG2000 with a ratio of 5/1.

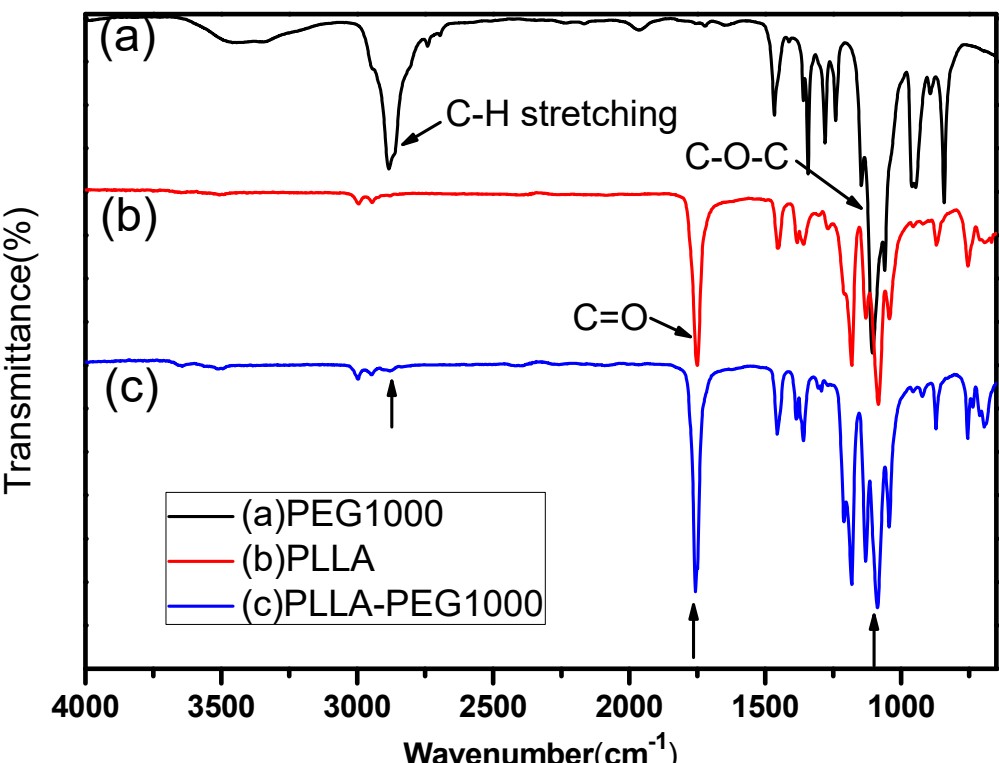

**Figure 2.** The FTIR spectra of PEG1000, PLLA, and PLLA-PEG1000 copolymers. (**a**) PEG1000 (**b**) PLLA (**c**) PLLA-PEG1000.

Using gel permeation chromatography (GPC) measured the average molecular weight of the PLLA-PEG copolymer; each assay was made three times, and then the average value was taken. The average molecular weight (Mw) of PLLA-PEG 200, 1000, and 2000 was 39,148, 27,604, and 28,255, respectively. The average molecular weight (Mn) numbers were 23,915, 18,561, and 18,612; the polydispersity index (Mw/Mn) was 1.638, 1.480, and 1.518. The molecular lengths of the series of the copolymer was distributed relatively evenly. The determining result demonstrated that the molecular weight of PLLA-PEG200 was higher than PLLA-PEG1000 and PLLA-PEG2000; the likely cause was that the molecular length of PEG200 was shorter, the steric effect was minor, and the terminal groups (-OH) made polymerizing easier among molecules, so that the molecular weight of PLLA-PEG200 was higher. The PLLA-PEG copolymers were stable in mechanical strength, translucent, and flexible materials.

*3.2. X-ray Diffraction of the PLLA-PEG/o-MMT Nanocomposites*

The nanocomposite material preparation for this study involved using the solution intercalation technique, which mixed organic matter and inorganic matter to composite materials. The matrix was PLLA-PEG copolymers, and the reinforcement was organically modified montmorillonite (o-MMT). In an attempt to let PLLA-PEG copolymers intercalate into the montmorillonite through the solution intercalation technique, the montmorillonite of inorganic matter should be nano-organized ahead. The amount of hydrophobic agents

was the key to nanosized montmorillonite; therefore, the size of the hydrophobic agent appeared to count for much. The bigger size caused unequal dispersion and then led to restraining the effectiveness. The main reason for organically modified montmorillonite was to improve its hydrophobic by modifying montmorillonite (because matrix PLLA-PEG copolymers were hydrophobic). By doing so, the matrix (PLLA-PEG) and reinforcement (o-MMT) were hydrophobic, and the o-MMT size reached "nano-standard"; thus, PLLA-PEG copolymers could intercalate into montmorillonite through the solution intercalation technique. In other words, the polymer intercalated into the silicate layers of montmorillonite via the assistance of the solution.

Figure 3 shows the X-ray diffraction patterns of modified organized montmorillonite (o-MMT) and a series of PLLA-PEG copolymer nanocomposites, adding different proportions of o-MMT. Figure 3a was the X-ray diffraction pattern of the modified o-MMT. The diffraction peak $2\theta$ of aspect d (001) was about $4°$, it meant that the distance between layers of the lamellar structure of montmorillonite was about 22 Å. Figure 3b–j were the X-ray diffraction patterns of the modified o-MMT with proportions of 0.5 wt%, 1 wt%, and 1.5 wt%, respectively, mixed with PLLA-PEG200, PEG1000, and PEG2000 copolymer nanocomposites.

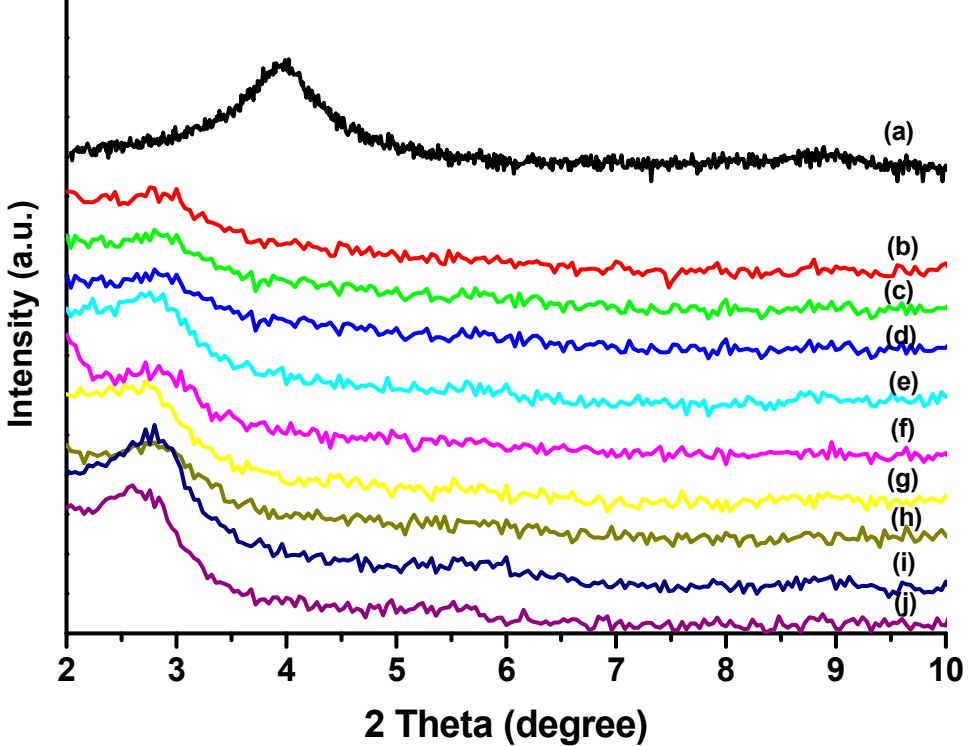

**Figure 3.** Curve (**a**) was the X-ray diffraction pattern of the modified o-MMT. Curves (**b**–**d**) were the X-ray diffraction patterns of PLLA-PEG200/o-MMT; the amount of o-MMT was 0.5 wt%, 1 wt%, and 1.5 wt%, respectively. Curves (**e**–**g**) were the X-ray diffraction patterns of PLLA-PEG1000/o-MMT; the amount of o-MMT was 0.5 wt%, 1 wt%, and 1.5 wt%, respectively. Curves (**h**–**j**) were the X-ray diffraction patterns of PLLA-PEG2000/o-MMT; the amount of o-MMT was 0.5 wt%, 1 wt%, and 1.5 wt% respectively.

Figure 3b–d, show the crystallization peak of o-MMT, in which $4°$ of aspect $d_{(001)}$ was invisible, meaning that the distance between layers of the lamellar structure of montmorillonite was expanded to a pretty significant degree. At this time, silicate's lamellar structure of o-MMT was utterly destroyed and then scattered over the PLLA-PEG matrix evenly to reach complete exfoliation. Figure 3e–g were X-ray diffraction patterns of the PLLA-PEG1000 copolymer, the original crystallization peak of o-MMT, in which $4°$ of aspect $d_{(001)}$ disappeared; however, there was a new crystallization peak appearing at

2.8° in 2θ, meaning that the interlayer distance of o-MMT expanded to 31.5 Å (3.15 nm). The result represented that the molecular length of PLLA-PEG1000 was longer, the steric effect was more, which was disadvantaged into the layers of o-MMT, so that o-MMT displayed a more obvious intercalated structure. Figure 3h–j were X-ray diffraction patterns of PLLA-PEG2000 copolymers, and the results exhibit the same phenomenon as that of PLLA-PEG1000 copolymers.

### 3.3. The Morphology of the PLLA-PEG/o-MMT Nanocomposites

Figure 4 was magnified 25,000 times the TEM of PLLA-PEG1000/0.5% o-MMT nanocomposites. The dispersed condition over the lamellar structure of o-MMT mostly displayed exfoliation, but it still had an intercalated form. Figure 5 was magnified 100,000 times TEM of PLLA-PEG1000/1.0% o-MMT nanocomposites. The intercalated structure could be seen clearly under high magnification. It matched the result of the X-ray diffraction pattern analysis, which verified that PLLA-PEG1000/1.0% o-MMT nanocomposite material owned the combination of exfoliation and intercalation. The dispersion morphology of PLLA-PEG1000/1.5% o-MMT and PLLA-PEG2000/1.5% o-MMT nanocomposites was observed, and the exfoliation was much reduced by increasing the amount of o-MMT in the nanocomposites. In summary, PLLA-PEG1000 and PLLA-PEG2000 copolymer/o-MMT nanocomposites had both exfoliation and intercalation structure types.

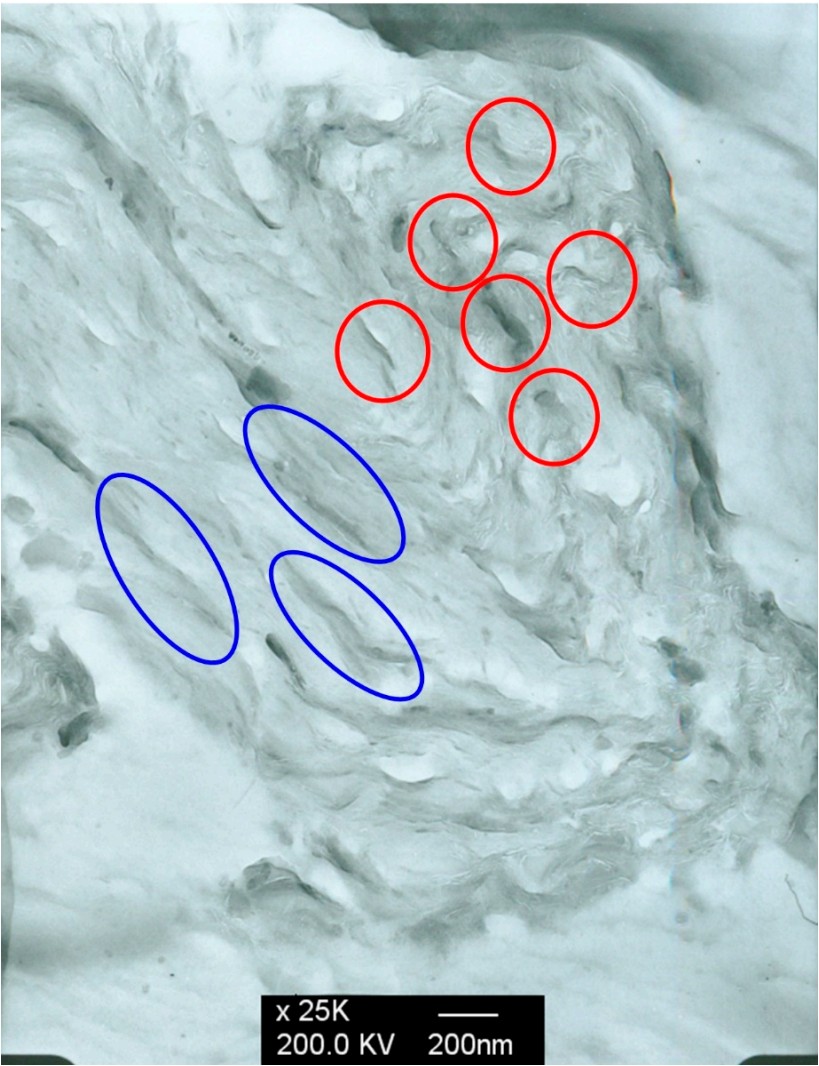

**Figure 4.** The TEM of PLLA-PEG1000/0.5% o-MMT nanocomposite material (magnification: ×25,000). The red ring shows the exfoliation; the blue ring shows the intercalated structure.

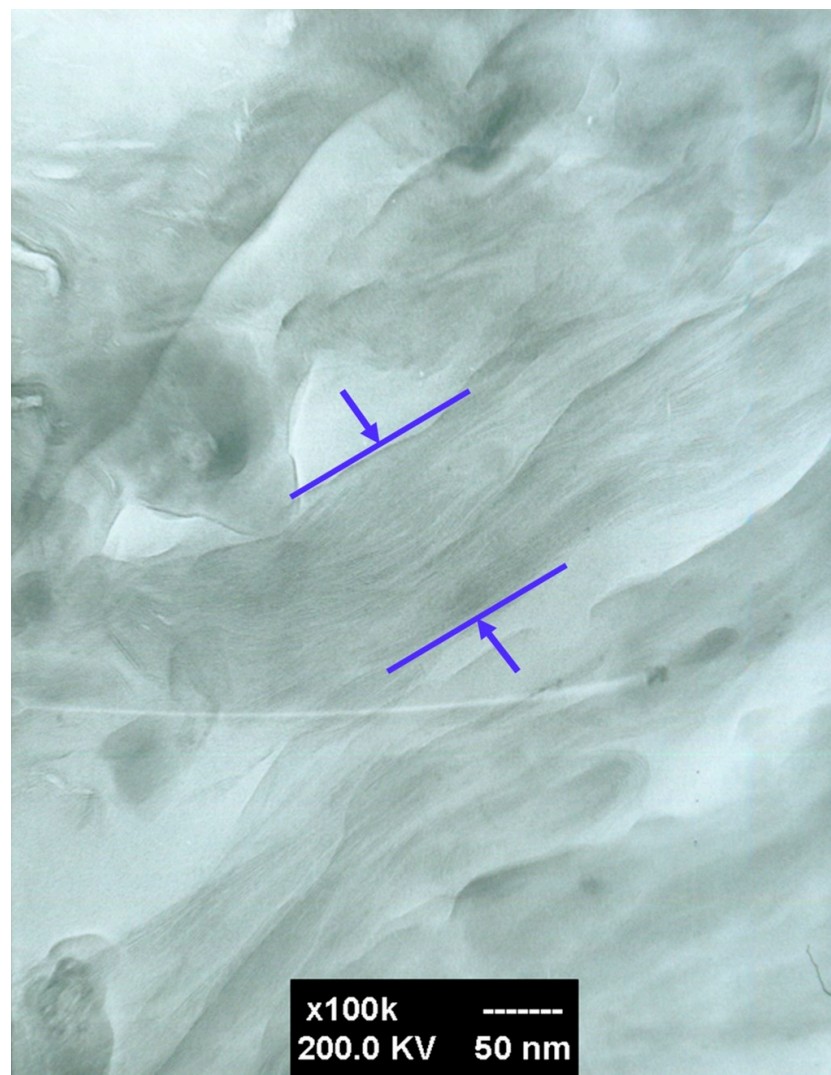

**Figure 5.** The TEM of PLLA-PEG1000/1.0% o-MMT nanocomposites (magnification: ×100,000). The blue marking shows an intercalated structure and a silicate layer of o-MMT with a thickness of about 100 nm.

*3.4. Non-Isothermal Crystallization of the PLLA-PEG/o-MMT Nanocomposites*

Differential scanning calorimetry compared the change in temperature of sample and thermal inertia material. The reference was measured as a function record of sample temperature, inertia material temperature, and stove temperature through invariable heating or cooling rate. The relative change of the sample temperature and inertia material temperature is caused by the sample's heat absorption transforming, endothermic reaction, heat release transforming, or exothermic reaction, such as phase change, fusion, crystal structure change, etc. In general, phase transforming created an endothermic reaction, and the crystallization reaction led to an exothermic reaction.

The PLLA-PEG/o-MMT nanocomposites with various o-MMT amounts (0, 0.5, 1.0, and 1.5 wt%) proceeded with thermal crystallization analyses. The procedures entirely involved "melt": heating the assay up 190 °C to completely melt; reducing the temperature by a cooling rate of 1 °C/min to generate crystallization. Figure 6 is the temperature reducing curve of the PLLA-PEG200 copolymer and nanocomposites, adding different proportions of o-MMT under a non-isothermal environment to proceed with crystallization. Table 1 shows the crystallization half time ($t_{1/2}$), enthalpies of crystallization ($\Delta Hc$), and the crystallization time list of the PLLA-PEG copolymer and nanocomposites, adding different proportions of o-MMT.

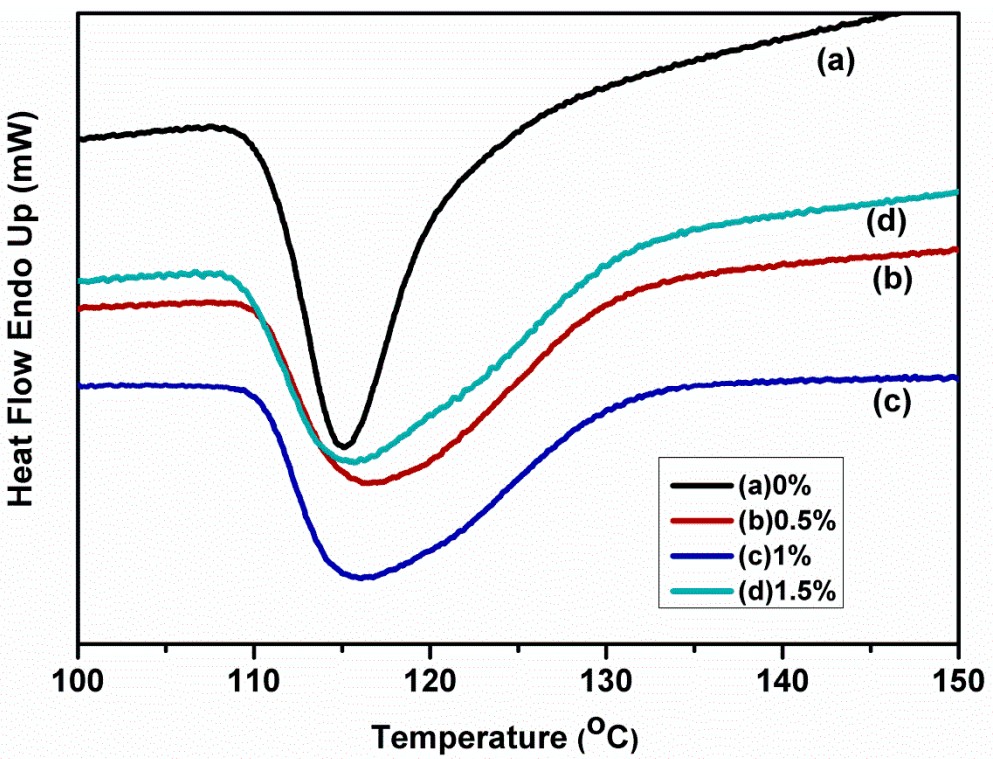

**Figure 6.** The DSC diagram of PLLA-PEG200 with different amounts of o-MMT nanocomposite material. The crystallization curves were under non-isothermal crystallization. The amount of o-MMT was (**a**) 0 wt%, (**b**) 0.5 wt%, (**c**) 1.0 wt%, and (**d**) 1.5 wt%, respectively.

**Table 1.** The $t_{1/2}$, $\Delta H_c$, and crystallization time list of the PLLA-PEG copolymer and the nanocomposites, adding different amounts of o-MMT.

| Ratio | PLLA-PEG200 | | | PLLA-PEG1000 | | | PLLA-PEG2000 | | |
|---|---|---|---|---|---|---|---|---|---|
| | $t_{1/2}$ (min) | $\Delta H_c$ (J/g) | Time (min) | $t_{1/2}$ (min) | $\Delta H_c$ (J/g) | Time (min) | $t_{1/2}$ (min) | $\Delta H_c$ (J/g) | Time (min) |
| 0% | 11.52 | 38.97 | 23.66 | 11.70 | 42.97 | 21.32 | 12.26 | 40.80 | 19.90 |
| 0.5% | 11.62 | 39.21 | 27.00 | 11.71 | 46.19 | 21.75 | 12.36 | 39.73 | 25.41 |
| 1% | 11.61 | 43.61 | 32.91 | 12.18 | 42.93 | 30.21 | 12.49 | 38.75 | 25.22 |
| 1.5% | 11.57 | 40.12 | 29.23 | 12.25 | 42.01 | 29.14 | 12.60 | 40.19 | 21.19 |

Table 1 shows the $t_{1/2}$ changes of the PLLA-PEG copolymer nanocomposites. The $t_{1/2}$ of PLLA-PEG200 did not have a remarked increase along with increasing o-MMT. However, the results for PLLA-PEG1000 and PLLA-PEG2000 are different from those of PLLA-PEG200, in which $t_{1/2}$ increases with the increasing molecular weight of the PEG segments. In addition, after adding o-MMT, the $\Delta Hc$ value of PLLA-PEG200 nanocomposites is higher than that without o-MMT, and there is no change in the $\Delta Hc$ value of PLLA-PEG1000 nanocomposites. However, the $\Delta Hc$ value of PLLA-PEG2000 nanocomposites became lower after adding o-MMT. From these results, we infer that o-MMT has a better dispersion in PLLA-PEG200, so the $\Delta Hc$ value of PLLA-PEG200/o-MMT is higher; therefore, the sizes of PEG segments affect the $\Delta Hc$ value of the nanocomposites. According to the research by Cheng et al., in low molecular weight polyethylene oxide (PEO) fractions, nonintegral folded chain (NIF) crystals can be found during the isothermal crystallization process. These crystals grow, first at a transient state, and integral-folding chain (IF) crystals form later through an isothermal thickening or thinning process [36]. Our study also found that when the molecular weight (200) of PEO in the PLLA-PEG co-polymer is lower, it is easier to produce more integral-folded chain (IF) crystals, even with a higher proportion of o-MMT. However, when the molecular weight (1000) of the PEO is large, even a higher isothermal crystallization temperature, coupled with the presence of a higher proportion of

o-MMT, will not be conducive to the conversion of nonintegral folded chain (NIF) crystals into the integral-folded chain (IF) crystals, making the number of integral-folded chain (IF) crystals minor. This proposition can explain why PEG molecular weight (1000) in o-MMT content is 0% and 0.5%, and there is a higher ΔHc. In terms of crystallization time, the non-isothermal crystallization time of PLLA-PEG/o-MMT nanocomposites, after adding o-MMT, is proportional to the area of the cooling curve. Therefore, based on Figure 6, the crystallization time of PLLA-PEG copolymer with o-MMT addition was longer than that without o-MMT addition. From the results, it is clear that the addition of o-MMT decreases the crystallization rate of PLLA-PEG copolymer; the longer the PEG segments length, the faster the crystallization rate at the same amount of o-MMT.

Figure 7 shows the thermal melting curves of PLLA-PEG 200 mixed with different proportions of PLLA-PEG 200/o-MMT nanocomposite materials under non-isothermal melting crystallization. The melting temperature peak was "triplet", based on Figure 7. Therefore, it could be presumed that it was a recrystallization phenomenon as part of crystallization melted under the heating process. Furthermore, the figure shows that the Tm was reducing, along with the amount of o-MMT added. Thus, adding o-MMT would affect crystallization morphology.

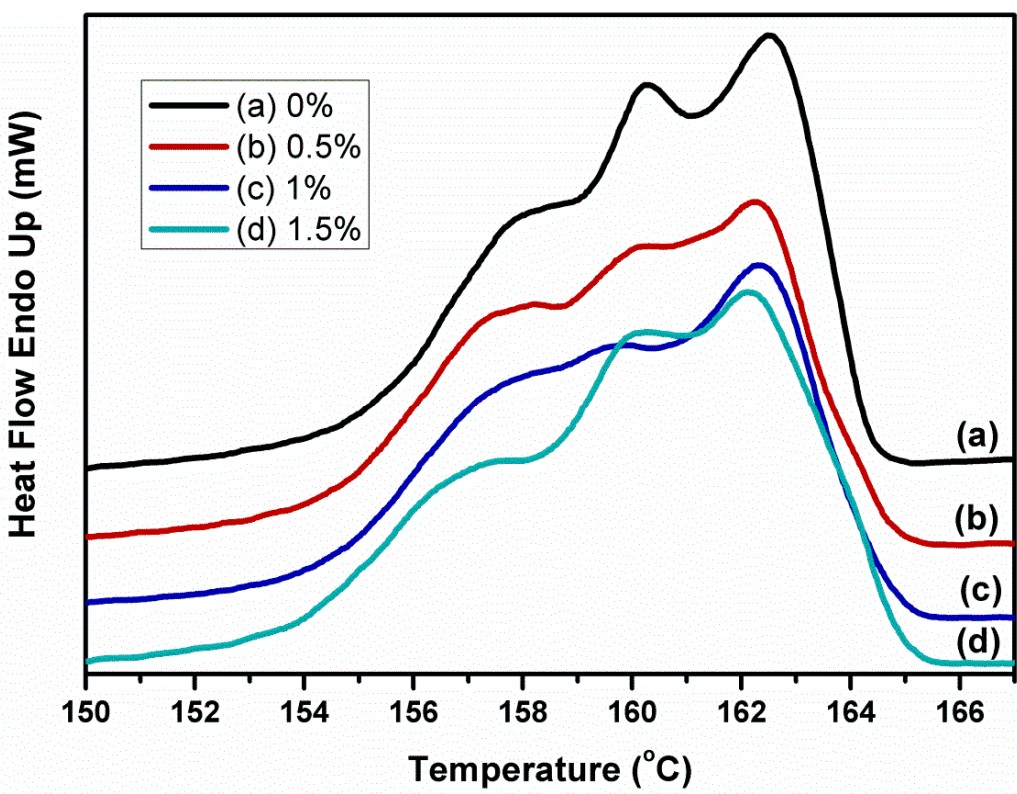

**Figure 7.** The DSC diagram of PLLA-PEG200 copolymer with adding different amounts of o-MMT. These DSC melting curves resulted from twice heating under non-isothermal melting crystallization. The amount of o-MMT was (**a**) 0 wt%, (**b**) 0.5 wt%, (**c**) 1 wt%, and (**d**) 1.5 wt%, respectively.

To summarize the results of Table 2—the melting temperature of PLLA-PEG200 and PLLA-PEG1000 did not change obviously for the copolymers added o-MMT. However, when the amount of o-MMT was 1.5 wt%, the melting temperature change of its PLLA-PEG2000 was significant, and the $T_m$ was lower than that without the addition of o-MMT by about 2 °C. In respect to the variation of enthalpies of fusion (ΔH$_f$), only the ΔH$_f$ of PLLA-PEG2000 copolymers were higher than that of PLLA-PEG2000 without o-MMT addition, but with ΔH$_f$ of PLLA-PEG200 and PLLA-PEG1000 copolymers, some of them were higher than non-adding o-MMT and some were lower; the results were irregular. It is well documented that the higher the ΔH$_f$, the greater the crystallinity [21]. As shown in

Table 2, the PLLA-PEG1000 copolymer with 0.5 wt% o-MMT had the most increased $\Delta H_f$ among a series of PLLA-PEG nanocomposites, so its crystallinity should also be the highest. The above results show that the best degree of crystallinity of PLLA-PEG copolymers was related to the PEG segment length and the amount of o-MMT. In other words, the proper collocation of these two parameters could produce higher crystallinity of nanocomposites.

**Table 2.** The isothermal crystallization peak and $\Delta H_f$ list of PLLA-PEG copolymer and nanocomposites added different amounts of o-MMT.

| Ratio | PLLA-PEG200 | | PLLA-PEG1000 | | PLLA-PEG2000 | |
|---|---|---|---|---|---|---|
| | Peak (°C) | $\Delta H_f$ (J/g) | Peak (°C) | $\Delta H_f$ (J/g) | Peak (°C) | $\Delta H_f$ (J/g) |
| 0% | 162.5 | 45.45 | 167.6 | 46.43 | 163.2 | 40.76 |
| 0.5% | 162.2 | 43.22 | 167.4 | 46.87 | 162.1 | 42.99 |
| 1% | 162.3 | 46.34 | 167.8 | 44.65 | 162.1 | 41.52 |
| 1.5% | 162.1 | 44.62 | 167.5 | 44.49 | 161.5 | 44.28 |

*3.5. Isothermal Crystallization of the PLLA-PEG/o-MMT Nanocomposites*

Figure 8 shows the isothermal crystallization curves of the PLLA-PEG200 copolymer, adding different amounts of o-MMT under different temperatures ($T_1$, $T_2$, $T_3$, $T_4$, and $T_5$). It showed that the crystallization times of both the PLLA-PEG200 copolymer and PLLA-PEG200/o-MMT nanocomposites were shorter when the isothermal crystallization temperature was lower. It indicated that the crystallization rate was faster under low temperatures because the nucleation rate was faster while under low temperature, which benefited from raising the bulk crystallization rate. Table 3 summarizes Figure 8, including crystallization half time ($t_{1/2}$), enthalpies of crystallization ($\Delta Hc$), and crystallization time. Table 3 shows that, no matter what temperature proceeded with isothermal crystallization after adding o-MMT, the $t_{1/2}$ and crystallization time of PLLA-PEG200/o-MMT nanocomposites were longer than the PLLA-PEG200 copolymer crystallization time; it meant that o-MMT decreased the PLLA-PEG200 copolymer crystallization rate. The $t_{1/2}$ of the PLLA-PEG200/o-MMT nanocomposite material, by the amount of o-MMT from high to low, was 1.5 wt% > 0.5 wt% > 1.0 wt%; however, the crystallization time, from long to short, was 1.0 wt% > 1.5 wt% > 0.5 wt%. Interestingly, the $t_{1/2}$ of PLLA-PEG200/1.0 wt% o-MMT nanocomposites was the shortest, but the crystallization time was the longest. It was assumed that the whole crystallization rate of materials was based on nucleation and growth; $t_{1/2}$ was the time needed for 50% crystallization and the key point was the speed of nucleation rate. This result shows that the nucleation rate of PLLA-PEG200/1.0 wt% o-MMT nanocomposites was the fastest initially, but its crystallization rate turned out slower at the end. The $\Delta Hc$ of PLLA-PEG200 copolymer was reduced by adding o-MMT, the lowest one was adding 1% o-MMT. There was a similar situation in the isothermal crystallization experiment of PLLA-PEG1000: the $\Delta Hc$ of PLLA-PEG1000, without adding o-MMT, was comparatively higher than those that added o-MMT. It illustrated that added o-MMT could reduce $\Delta Hc$, and it influenced the crystal nucleating crystallization rate. Okamoto et al. once used polarized light microscopy, light scattering, differential scanning calorimetry, and wide-angle X-ray diffraction to discuss and analyze crystallization kinetics. The PLA is crystallized at an isothermal temperature, and the pure PLA crystals are typically orthorhombic crystals. However, PLA is added to MMT, the nanocomposites are crystallized in a defect-ridden crystalline form, because this unstable growth of crystallites of PLA, in the presence of MMT particles, may be due to the intercalation of PLA chains into the silicate layers of MMT. As a result, the MMT silicate layer becomes a nucleating agent in the material, causing the overall crystallization rate and spherical structure of pure PLA to be strongly affected [35,37]. Similarly, in our study, it was found that when the MMT silicate layer was uniformly dispersed in the PLLA-PEG copolymer, a nucleating agent was formed, and the crystallization rate and the regularity of the crystals changed with the increase of the MMT content, which further affected the $\Delta Hc$ and $\Delta H_f$.

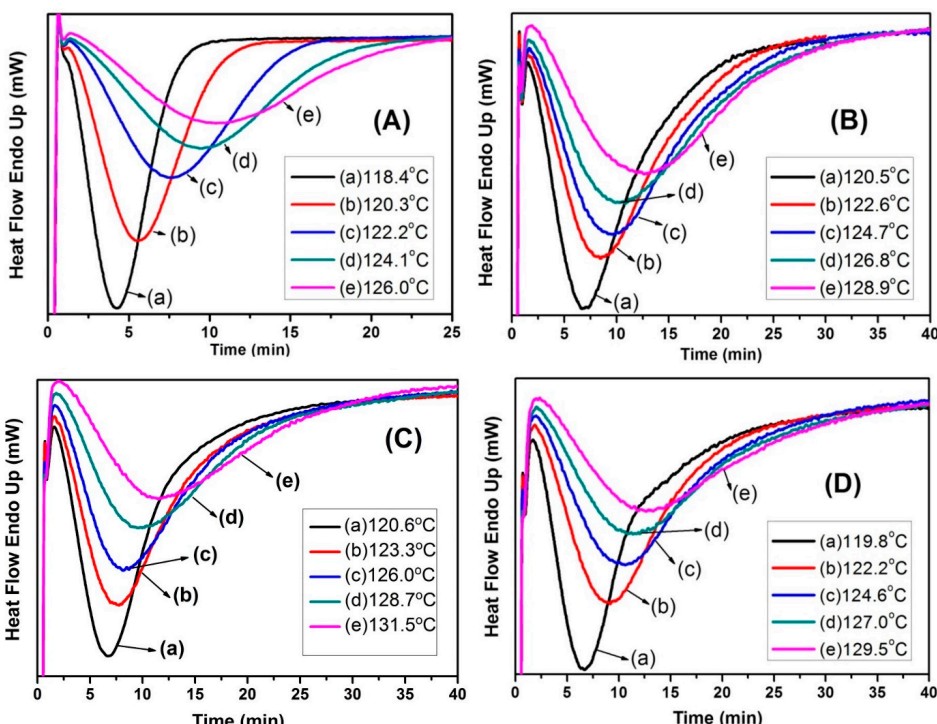

**Figure 8.** The DSC diagram of the PLLA-PEG200 copolymer, adding different amounts of o-MMT. The crystallization curves were under isothermal crystallization. The amount of o-MMT was (**A**) 0%, (**B**) 0.5%, (**C**) 1.0%, and (**D**) 1.5%, respectively.

**Table 3.** The $t_{1/2}$, $\Delta H_c$, and crystallization time list of PLLA-PEG200 copolymer and PLLA-PEG200/o-MMT nanocomposites under isothermal crystallization.

| Temp. | 0% | | | 0.5% | | |
|---|---|---|---|---|---|---|
| | $t_{1/2}$ (min) | $\Delta H_c$ (J/g) | Time (min) | $t_{1/2}$ (min) | $\Delta H_c$ (J/g) | Time (min) |
| $T_1$ | 4.23 | 37.14 | 8.92 | 6.65 | 34.86 | 25.63 |
| $T_2$ | 5.55 | 39.48 | 12.30 | 8.43 | 35.82 | 28.51 |
| $T_3$ | 7.47 | 40.69 | 17.59 | 9.45 | 37.97 | 33.90 |
| $T_4$ | 9.52 | 41.32 | 24.53 | 10.80 | 38.87 | 38.19 |
| $T_5$ | 10.38 | 42.06 | 31.31 | 12.58 | 40.61 | 54.46 |

| Temp. | 1% | | | 1.5% | | |
|---|---|---|---|---|---|---|
| | $t_{1/2}$ (min) | $\Delta H_c$ (J/g) | Time (min) | $t_{1/2}$ (min) | $\Delta H_c$ (J/g) | Time (min) |
| $T_1$ | 6.60 | 30.12 | 36.36 | 6.65 | 32.98 | 32.66 |
| $T_2$ | 7.77 | 31.60 | 41.19 | 9.13 | 35.40 | 37.05 |
| $T_3$ | 8.12 | 32.55 | 47.87 | 10.60 | 36.67 | 42.58 |
| $T_4$ | 9.57 | 34.19 | 57.79 | 11.38 | 37.82 | 55.50 |
| $T_5$ | 11.45 | 35.90 | 79.71 | 12.58 | 38.93 | 67.35 |

Figure 9 shows twice the heating melting curves of the PLLA-PEG200 copolymer, adding different proportions of o-MMT under different temperatures ($T_1$, $T_2$, $T_3$, $T_4$, and $T_5$), proceeding with isothermal crystallization. The figure reveals that PLLA-PEG200 copolymers were "double-melting peak" because they contained two different macromolecules. However, "twice-heating" melting produced triple-melting behavior under lower temperatures ($T_1$, $T_2$, $T_3$, $T_4$, and $T_5$) while proceeding with isothermal crystallization, after adding o-MMT. In particular, PLLA-PEG200/0.5 wt% o-MMT and PLLA-PEG200/1.5 wt% o-MMT were the most significant. The $Tm_3$ of the highest temperature, at about 163.5 °C, it was assumed that parts of the crystallization melted and then crystallized again, along with the DSC heating procedure. The peak value was heightened when the amount of

o-MMT increased, but it lowered while crystallization temperature increased. When the temperature was up to $T_5$, it is found that $Tm_3$ disappeared because the crystallization rate was comparably much slower under high temperatures, which made the crystalline structure stop crystallizing. Furthermore, Table 4 indicates that $\Delta H_f$ of the PLLA-PEG200 copolymer, after adding o-MMT, was lower than without adding it, and 1.0 wt% o-MMT of PLLA-PEG200/o-MMT nanocomposites could get the lowest $\Delta H_f$, which meant its crystallinity was the smallest. In summary, the addition of o-MMT to PLLA-PEG copolymers will reduce its crystallization rate and crystallinity, mainly because the silicate layer of the o-MMT is dispersed in the PLLA-PEG copolymers; thus, forming more grains, but these grains have different growth rates in other isothermal crystallization processes, as evidenced by the changes in $t_{1/2}$, $\Delta Hc$, $\Delta H_f$, and Tm of these PLLA-PEG nanocomposites.

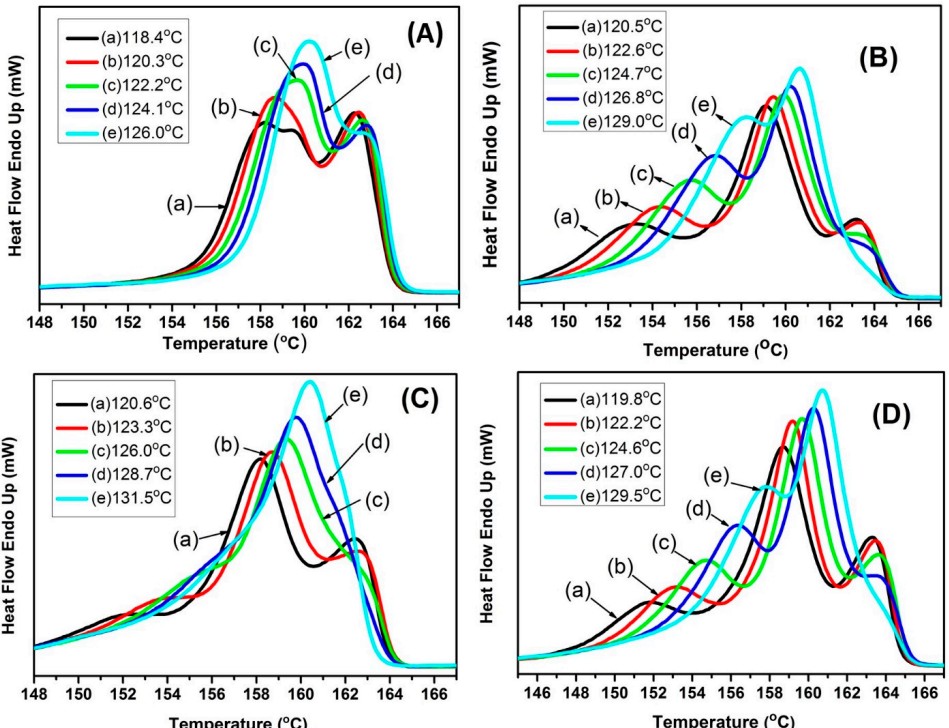

**Figure 9.** The DSC diagram of PLLA-PEG200 copolymer with adding different amounts of o-MMT. These DSC melting curves resulted from twice-heating under different temperatures, proceeding with isothermal crystallization (a) $T_1$ (b) $T_2$ (c) $T_3$ (d) $T_4$ (e) $T_5$. The amount of o-MMT was (**A**) 0%, (**B**) 0.5%, (**C**) 1.0%, and (**D**) 1.5%, respectively.

Figure 10 shows the corresponding melting point diagram between $T_m$ and $T_c$ of PLLA-PEG copolymers and PLLA-PEG/o-MMT nanocomposites under different isothermal crystallization temperatures. We could realize that the relationship between $T_m$ and $T_c$ were almost in direct proportional linearity (either PLLA-PEG copolymer or PLLA-PEG/o-MMT nanocomposites). To paraphrase—the higher isothermal crystallization temperature will result in higher melting point for the PLLA-PEG copolymer and PLLA-PEG/o-MMT nanocomposites. Because the crystallization rate was slower while crystallization temperature was higher, this made chain folding more orderly, and then $T_m$ was higher. By comparing (A), (B), and (C) in Figure 10, we understand that $T_m$ decreased after adding o-MMT in different molecular weights of the PLLA-PEG copolymer, but the effect levels were not the same. For example, when the isothermal crystallization temperature is 126 °C, the addition amount of 0.5 wt% o-MMT will decrease the $T_m$ of PLLA-PEG2000 (2.5 °C) > PLLA-PEG1000 (1.5 °C) > PLLA-PEG200 (0.8 °C). For o-MMT, the $T_m$ of PLLA-PEG copolymer was related to the PEG segment length. As the molecular chain length was shorter, it was easier to intercalate the layers between o-MMT, which less affected the melting point ($T_m$). Furthermore, in Figure 10A, it is evident that the $T_m$ variation values

of PLLA-PEG200 nanocomposites with three different ratios of o-MMT added were not significant. This is because the silicate layer of o-MMT was uniformly dispersed in the PLLA-PEG200 nanocomposite.

**Table 4.** The melting peak and melting enthalpy ($\Delta H_f$) list of the twice-heating curve under isothermal crystallization on PLLA-PEG200 copolymer and PLLA-PEG200/o-MMT nanocomposites.

| Temp. | 0% | | 0.5% | | 1.0% | | 1.5% | |
|---|---|---|---|---|---|---|---|---|
| | Peak (°C) | $\Delta H_f$ (J/g) | Peak (°C) | $\Delta H_f$ (J/g) | Peak (°C) | $\Delta H_f$ (J/g) | Peak (°C) | $\Delta H_f$ (J/g) |
| $T_1$ | 158.1 | 44.62 | 159.1 | 40.63 | 158.2 | 39.16 | 158.7 | 40.93 |
| $T_2$ | 158.6 | 44.98 | 159.4 | 41.53 | 158.7 | 40.02 | 159.2 | 41.81 |
| $T_3$ | 159.7 | 45.26 | 159.9 | 43.40 | 159.4 | 41.83 | 159.7 | 43.42 |
| $T_4$ | 159.9 | 45.38 | 160.2 | 44.12 | 159.8 | 43.10 | 160.3 | 45.22 |
| $T_5$ | 160.2 | 46.26 | 160.7 | 45.26 | 160.4 | 44.47 | 160.7 | 46.64 |

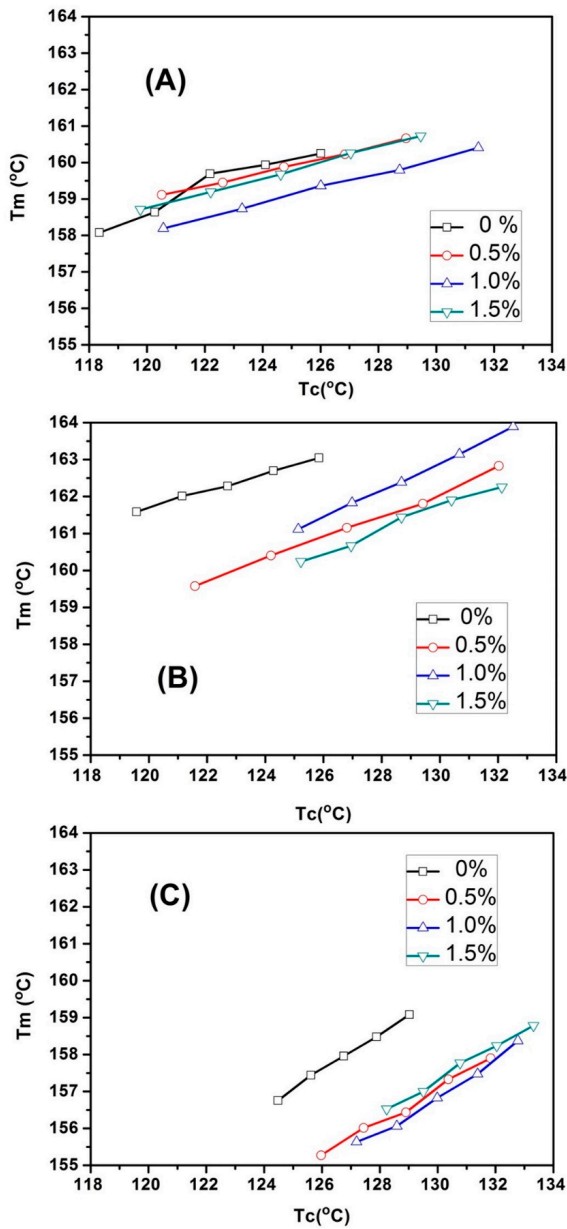

**Figure 10.** The corresponding diagram between $T_m$ and $T_c$ of PLLA-PEG copolymer and PLLA-PEG/o-MMT nanocomposites. (**A**) PLLA-PEG200, (**B**) PLLA-PEG1000, and (**C**) PLLA-PEG2000 copolymer.

## 4. Conclusions

Disposable plastic materials have caused serious environmental pollution, landscape destruction, and marine ecological hazards. Therefore, the development and manufacturing of biodegradable plastic materials have become crucial in this era. In this study, three PLLA copolymers–PLLA-PEG200, PLLA-PEG1000, and PLLA-PEG2000 were successfully synthesized, and a series of non-toxic PLLA-PEG/o-MMT nanocomposites with organically modified montmorillonite were successfully prepared. X-ray diffraction patterns and a transmission electron microscope showed that the molecular weight of PEG in the PLLA-PEG copolymers would affect the dispersion state of the silicate layer of o-MMT. The o-MMT in the PLLA-PEG200 copolymers mainly exhibited the exfoliation pattern, but in the PLLA-PEG1000 and PLLA-PEG2000 copolymers, the o-MMT exhibited exfoliation and intercalation.

This study shows that, under isothermal crystallization conditions, the lower the crystallization temperature of the PLLA-PEG/o-MMT nanocomposite, the crystal defects could occur, so the lower the $\Delta H_c$, $\Delta H_f$, and $T_m$. This phenomenon indicates that the crystallinity is lower. The crystallization of the PLLA-PEG/o-MMT nanocomposites was affected by the content of the o-MMT and the molecular weight of PEG of the components. Among a series of PLLA-PEG/o-MMT nanocomposites, the PLLA-PEG200/o-MMT nanocomposites with 1.0 wt% o-MMT have the lowest crystallinity, and the PLLA-PEG1000 nanocomposites with 0.5 wt% o-MMT have the highest crystallinity. The higher the crystallinity of the material, the higher the hardness, and the higher the thermal stability, but the material may become brittle. As a result of this study, it is possible to adjust the crystallinity of materials appropriately using these nanocomposites, which can be referenced for further investigation.

Furthermore, it is confirmed that adding o-MMT could reduce the melting point of PLLA-PEG copolymers; the smaller the molecular weight of the PEG chain, the smaller the effect on the decrease of $T_m$. Whether it is PLLA-PEG copolymers or PLLA-PEG/o-MMT nanocomposites, the relationship between $T_m$ and $T_c$ is almost linearly proportional. This result proves that the silicate layers of montmorillonite should be relatively uniformly dispersed in the PLLA-PEG copolymers.

To conclude, it was proven that the PLLA-PEG/o-MMT nanocomposites, by adjusting the additional amount of o-MMT and the length of the PEG segment, have more advantageous crystallinity and physical properties compared to the PLLA-PEG copolymers. The mechanically more stable biodegradable materials could be produced, which could replace disposable plastic materials and be used for biomedical materials, such as materials for fixing fracture sites, to reduce the risk and pain of secondary operations in the future. It is believed that PLLA-PEG/o-MMT nanocomposites are promising materials for biomedical science and the sustainable development of the earth.

**Author Contributions:** J.-J.H. and H.-J.L. conceived of and designed the experiments; W.-H.C. and C.-C.C. prepared the samples and performed the experiments; S.-M.H. and J.-J.H. analyzed the data; W.-Y.L., J.-J.H. and S.-M.H. contributed to the writing and revising of the paper. All authors have read and agreed to the published version of the manuscript.

**Funding:** The authors would like to thank the National Science Council of Taiwan for financially supporting this research under Contract No. NSC 99-2622-E-238-015-cc3.

**Acknowledgments:** The authors would like to thank the Center for Precious Instruments of National Taiwan University for their assistance in NMR testing and the Center for Precious Instruments of Chung Yuan Christian University for their help in TEM sample testing.

**Conflicts of Interest:** The authors declare no conflict of interest.

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
