# Peer review of "The Synthesis of Biodegradable Poly(L-Lactic Acid)-Polyethylene Glycols Copolymer/Montmorillonite Nanocomposites and Analysis of the Crystallization Properties"

_minerals, doi:10.3390/min12010014_

Round 1

Reviewer 1 Report

The paper presents the research on the synthesis and crystallization characteristics of poly(L-lactic acid)-polyethylene glycols copolymer/montmorillonite nanocomposite materials. The current form’s presentation of methods and scientific results is satisfactory for publication in the Minerals journal. The minor and significant drawbacks to be addressed can be specified as follows:
1.    (i) Tittle: Materials ---> Materials. (ii) Polyethylene Glycols ---> polyethylene glycols. See the article for the other typos.
2.    “2.2. Lipophilic property treating of montmorillonite.” And “3.3. The Morphology of the PLLA-PEG/ o-MMT Nanocomposites” Was there any exfoliation of montmorillonite layers? While analyzing Figs 4 and 5, I am not convinced that what the authors write about exfoliation and intercalation is consistent with reality. What is the distance between layers and layer thickness for o-MMT.
3.    Fig. 8. Why are there other colours on the panel (c)? See also Fig. 9: (a) and (c) vs. (b) and (d). Assuming one way of marking the curves (colours) would facilitate the analysis of the graphs.
4.    Fig. 10. What was the repeatability of sample preparation? From the analysis of this Figure, it appears that the processing of nanocomposites is quite random - there are no trends. Without error analysis and determination of error bars, it’s hard to draw any conclusions. Unfortunately, the authors’ results confirm the “difficulty” of this type of composite material.

Author Response

We want to thank the reviewers for their suggestions on this paper, and we have also made changes to address the shortcomings that need improvement:

  1. We have carefully checked and corrected all the errors in the title of the paper and the typing of the text.
  2. We have also published 2.2, “preparation of the organophilic montmorillonite” in other journals, e.g., ref. 34, 35. Therefore, we have added in 3.2 that the interlayer distance can be expanded to 31.5 Å(3.15nm) after intercalation of PLLA/PEG-1000 and o-MMT (in the first paragraph of page 13 of the revised manuscript) and added in Figure 5. As a result, the thickness of the intercalated silicate layer is about 100 nm.
  3. Fig. 8 and Fig. 9 we have corrected them to the same color of the marked curves, which helps analyze the results.
  4. In Fig. 10, we modify (A)(B)(C) to the same X-coordinate range (118~134℃) and Y-coordinate range (155~164℃). In this way, we can see that the different lengths of the PEG segment will also cause changes in the melting point, resulting in different crystallization temperature ranges. The results show a linear trend, indicating that the PLLA/PEG nanocomposites' homogeneity is perfect.

Our revised paper is uploaded as an attachment. Please see the attachment.

Reviewer 2 Report

This manuscript cantains numerous English errors (e.g. the title itself,  in Abstract: The less molecular weight of poly(ethylene glycol), The nanocomposite materials were measured, From measurement of non-isothermal crystallization found, etc.), and therefore its merit can't be understood well.  English needs to be checked by a native speaker and an appropriate certificate provided.   

Author Response

We want to thank the reviewers for their suggestions on this paper. This paper also invites native speakers with a professional background to assist in revising the grammar, vocabulary and rewriting some sentences to make the text clear so that readers can fully understand the value and importance of this paper.

Reviewer 3 Report

Introduction: Some important research results in this area should be discussed in the section of introduction. Please provide a solid background and progress to the readers regarding the current state-of-knowledge on this topic, and tell readers what the highlights of your manuscript are compared with previous progresses about PLLA-PEG copolymers (+/- montmorillonite) nanocomposite materials. From the 45 references used in the introduction, please keep only the relevant literature data.

Materials and methods. This section is too long and should be more compacted, while keeping the most important information. This section is written too general as a kind of report, and not suitable for a scientific paper. Revise the synthesis section in a more “technical” way. Also, please standardize the details regarding all used equipment (such as name, model, manufacturer, measurement conditions, etc.) for measurements.

Results and Discussion. To increase the scientific value of the manuscript Authors should consider extension of the whole results section with comparison of obtained results with the results described in previous publications (such a large field).

Chapter Conclusions should be definitely rewritten. It must be extended, just give some specific results of your research.

English of the paper is rather good – in my opinion the language of the paper should be a little improved. I am asking for corrections by a native speaker. Once again: you research is valuable- please consider my comments to improve your paper.

The results obtained are interesting and promising, but poorly discussed compared to similar studies. The manuscript can be accepted for publication only after MAJOR corrections. If the manuscript will not be considerable revised, I will not recommend its publication.

Author Response

We want to thank the reviewers for their suggestions on this paper, and we have also made changes to address the shortcomings that need improvement:

  1. The references cited in this paper have been carefully read, summarized and analyzed, and the older or less relevant references have been removed. The weaknesses of PLA in applications, as well as the research and development of PLA copolymers and related applications, are explained and appropriate literature is cited.
  2. The highlights of this study are also added in the introduction. Firstly, the nanocomposite material made from PLLA, PEG, and clay is non-toxic and harmless to the human body during biodegradation. Secondly, it is expected that the PLLA-PEG nanocomposite with the addition of o-MMT can improve the crystallization property of the original PLLA and control the processing temperature. The effect of PEG molecular size on the morphology and related crystallization properties of PLLA-PEG nanocomposites is also discussed.
  3. We have revised the experimental methods and instrumental measurement conditions to conform to the format of a scientific paper. Still, some of the experimental procedures and measurement conditions may confuse the reader if too much is removed.
  4. In the Results and Discussion, we have corrected some unclear points to make this research paper easy to read and understand and organize the following key points of the study.
  • Mechanical strength: Originally, PLA is a brittle and inflexible material; it is easily broken by external force and is not resistant to impact. The PLLA-PEG copolymer synthesized in this research is a material with stable mechanical strength. (in the second paragraph of page 11 of the revised manuscript)
  • Novelty: The PLLA-PEG copolymer is synthesized by reacting different PEG segments (MW: 600, 1000, 2000) with lactide, and the copolymer is intercalated with montmorillonite to form a nanocomposite. At present, there are very few such studies in international journal papers. (in 3.2~3.3 of page 11~14, the revised manuscript)
  • Uniqueness: We are the first to publish in an international journal study on the morphological differences caused by the size of PEG molecular chains in PLLA-PEG copolymers and the TC and Tm of the PLLA-PEG nanocomposites. (There are additional instructions on page 20 in the revised manuscript)
  • Change in crystallinity: The addition of o-MMT to PLLA-PEG copolymers will reduce its crystallization rate and crystallinity, mainly because the silicate layer of the o-MMT is dispersed in the PLLA-PEG copolymers, thus forming more grains, but these grains have different growth rates in different isothermal crystallization processes, as evidenced by the changes in t1/2, â–³Hc, â–³Hf, and Tm of these PLLA-PEG nanocomposites. (in the second paragraph of page 19 of the revised manuscript)
  1. The concluding section has been rewritten to include the more specific findings of the study to highlight the importance of this research paper.

6. This paper also invites native speakers with a professional background to assist in revising the grammar, vocabulary and rewriting some sentences to make the text clear so that readers can fully understand the value and importance of this paper.

Our revised paper is uploaded as an attachment. Please see the attachment.

Round 2

Reviewer 1 Report

The authors have made a substantial improvement for this article. The manuscript can be accepted for publishment in the present form.

Author Response

We are very grateful to the reviewers for their encouragement and support, and we will continue our efforts in this research field.

Reviewer 2 Report

In  this work authors prepared through polycondensation poly(L-lactic acid)-(polyethylene glycol) (PLLA-PEG) biodegradable copolymer, and then fabricated a series of nanocomposites PLLA-PEG with organically modified montmorillonite(o-MMT). The intercalation / exfoliation of MMT layered structure was investigated by X-ray diffraction and transmission electron microscopy. Melting and crystallization behaviour of the nanocomposites were studied by DSC under non-isothermal and isothermal conditions. It has been found that incorporation of o-MMT affect the crystallization rate of PLLA-PEG copolymers.  The revised version of the manuscript has been considerable improved, both in terms of English and scientifically, however, some points still need to be adressed in more depth:

- Table 1- kindly explain  why deltaH value (0 and 0.5%) is the highest for middle PEG value (1000), and for 1% it is PEG200? Do you consider for PEG non-integral folding chain crystal (NIF) and  integral folding crystal formation [e.g. Macromolecules 1991, 24, 13, 3937–3944]?

- In regard to the statement „The results of the present study suggested that adding o-MMT could affect the crystallization rate of PLLA-PEG copolymers” please provide a discussion on the role of MMT as nucleation agent during (co)polymer crystallization [e.g. Macromolecules 2003, 36, 7126-7131].  

- English has been greatly improved, however, in Abstract there are still some sentences that need to be rewritten, e.g. The low molecular weight of poly(ethylene glycol) à The lower  molecular weight of poly(ethylene glycol); the nanocomposites were measured under non-isothermal crystallization à the nanocomposites were investigated under non-isothermal crystallization, etc.

Author Response

We want to thank the reviewer for providing relevant references to support this paper's arguments and make this paper more complete. We have supplemented and revised this paper with the suggestions made by the reviewer, and the explanatory notes are listed below:

(1) Regarding the question "Do you consider for PEG nonintegral-folding chain crystal (NIF) and integral-folding crystal (IF) formation [e.g., Macromolecules 1991, 24, 13, 3937-3944", we have read this reference carefully, and in our study, it is true that the difference in the proportions of NIF and IF crystalline forms may be due to the different lengths of PEO molecular chains, as we have explained in page 13 of the revised manuscript and cited the above paper as reference 36.

(2) Regarding the question, "Please provide a discussion on the role of MMT as nucleation agent during (co)polymer crystallization [e.g., Macromolecules 2003, 36, 7126-7131]". We have carefully read this reference. We agree that the uniform dispersion of the silicate layer of montmorillonite in PLLA-PEG copolymers would form a nucleation agent that would significantly affect the crystallization behavior of PLLA-PEG copolymers. Therefore, we added a note on page 16, 17 of the revised manuscript and cited the above paper as reference 37.

(3) We have corrected the sentences and words that reviewers pointed out in the abstract, and we have rewritten some sentences to make them more understandable. We used the "Track Changes" function for the corrections to help reviewers and editors.

(4) We also rechecked the paper for typing errors and fixed all the mistakes. We used the "Track Changes" function for the corrections to help reviewers and editors.

Reviewer 3 Report

The manuscript has been much strengthened by the additional data.  I appreciate the authors' effort. 

Author Response

(The authors gave the same response as above.)
